# Pain and Avoidance during and after Endodontic Therapy: The Role of Pain Anticipation and Self-Efficacy

**DOI:** 10.3390/ijerph19031399

**Published:** 2022-01-27

**Authors:** Noelia Santos-Puerta, Cecilia Peñacoba-Puente

**Affiliations:** 1Ph.D Program for Health Science, Rey Juan Carlos Doctoral College, C/Quintana, 2, 28008 Madrid, Spain; n.santos@alumnos.urjc.es; 2Department of Psychology, Rey Juan Carlos University, Avda. Atenas s/n, Alcorcón, 28922 Madrid, Spain

**Keywords:** avoidance, pain anticipation, self-efficacy, endodontic therapy

## Abstract

Background: Pain anticipation has been identified as a predictor of pain and avoidance with respect to endodontic therapy. Self-efficacy is also key to the development and maintenance of health behaviors and achieve patient adherence to treatment. However, the role of self-efficacy has not been studied yet in endodontic treatment. Methods: This study was conducted on 101 patients who needed root canal therapy. They had to fill a questionnaire before treatment registered pain anticipation and self-efficacy; during and after treatment were registered pain intensity and avoidance. Results: Pain anticipation explained pain during (Beta = 0.51, *t* = 5.82, *p* ≤ 0.001, [0.34, 0.69]) and after treatment (Beta = 0.38, *t* = 4.35, *p* ≤ 0.001, [0.21, 0.55]). Self-efficacy did not have an influence in pain values. Pain anticipation explained avoidance during (Beta = 0.51, *t* = 3.60, *p* ≤ 0.001, [0.23, 0.80]) and after treatment (Beta = 0.62, *t* = 4.29, *p* ≤ 0.001, [0.33, 0.91]). Self-efficacy had a significant role in avoidance during treatment (Beta = 0.12, *t* = 2.19, *p* ≤ 0.03, [0.01, 0.23]) with a strong moderation relationship between pain anticipation and avoidance when self-efficacy was medium (Beta = 0.44, *t* = 3.24, *p* = 0.002, [0.17, 0.72]) or high (Beta = 0.84, *t* = 3.5, *p* ≤ 0.001, [0.37, 1.33]). Self-efficacy was not significant respect to avoidance after treatment. Conclusions: Self-efficacy is an important variable in endodontic therapy due to their moderating effect between pain anticipation and avoidance behavior during the procedure. It is necessary to improve the results of root canal therapy and reduce patient’s avoidance in order to take into account this variable.

## 1. Introduction

For some patients, visiting the dentist can be an extremely stressful experience that may result in a perceived lack of coping skills [1]. This has been reported to be a major reason for patients to delay or cancel dental appointments and therefore for avoidance of dental care as a consequence. This may result in a significant deterioration of oral and dental health [2,3,4,5,6]. Thus, in order to provide the best level of care, it is necessary to identify a high number of patients in this situation [7,8,9]. In recent years, there has been an increasing interest in patient-centered care and the individualization of their treatment. This way, dentists could make better informed decisions about pain management and anesthesia techniques, based on the best evidence published. This is an ongoing research field, the objective of which is to reduce postoperative pain, improve procedure techniques, and materials and procedures such as the use of new rotary files [10,11] or irrigation solutions [12].

Despite these developments in endodontics, patients in general still have negative beliefs and misconceptions about endodontic treatments that might make them decide to avoid dental treatments due to the overestimated anticipated pain [13].

Therefore, these patients could benefit from therapies focused on reducing the sensation of lack of ability to confront these situations by studying their levels of their perceived self-efficacy. The term self-efficacy was introduced by Bandura in 1977 as part of his social cognitive theory [14]. Specifically, self-efficacy is defined as a person’s belief in his or her capability to successfully perform a particular task [14]. According to the self-efficacy theory and previous published research, self-efficacy influences how people feel, think, and act [15,16]. A low level of self-efficacy is linked to depression and anxiety. On the contrary, a high level of self-efficacy is associated with reduced levels of stress [17,18,19]. Moreover, people presenting higher levels of self-efficacy are more persistent in their actions and are committed with their goals even while facing difficulties [20].

Self-efficacy has been shown in the previous literature as one of the psychological constructs consistently associated with health, including areas such as health promotion, disease prevention, or adherence to treatment [21,22,23,24,25,26].

Particularly in dentistry, self-efficacy has been positively associated in periodontal patients in terms of the motivation to undergo a treatment, maintaining adequate brushing techniques, and use of dental floss [7,27,28,29,30,31,32]. Likewise, different research studies on patients having an orthodontic treatment, have shown that motivation at the beginning and continuity of treatment was positively associated with the level of self-efficacy [33,34,35]. With regard to Endodontics, self-efficacy has only been analyzed in relation to the performance of the professional performing root canal treatments [36,37,38], but, to our knowledge, the relationship of the patient’s self-efficacy with their health outcomes has not yet been studied.

Root canal treatments are usually identified as the most stressful dental procedures for patients as they anticipate more pain than what it is finally perceived [5,39,40,41,42]. Some studies suggest that the fear of suffering pain before having a root canal treatment, could modify the perception of pain during the procedure [43]. Besides pain, patients can be also concerned about the length of the procedure [8,44].

Therefore, self-efficacy could play a main role to determine how a patient is dealing with a treatment that might cause pain and how to individualize the treatment offered. For that reason, this is important for obtaining a better understanding of the situation so that psychological methods can be incorporated in the training of future dentists [45,46].

The main aim of this study was to analyze the perception of pain during endodontic therapy at two different time points: (a) during the procedure (cone fit X-ray) and (b) at the end of the treatment (appointment for final restoration). There are also two additional objectives: (1) to analyze the possible influence of pain anticipation and self-efficacy (as moderator role) in the perception of pain during and after the treatment, and (2) to assess the possible influence of pain anticipation and self-efficacy (as moderator role) in the avoidance during and after treatment.

## 2. Materials and Methods

### 2.1. Participants

The study was conducted in two different clinical settings: 59 participants were treated in an academic setting, University Dental Clinic, at the Health Sciences Faculty of Rey Juan Carlos University (Madrid, Spain) and 42 subjects were treated in a private dental practice (Ferrus and Bratos Dental Practice, Madrid, Spain). All root canal treatments were carried out by a single experienced endodontist (MSc Endodontics) using the same diagnostic tests, materials, and equipment (including a dental surgical operating microscope and the assistance of a qualified dental nurse at all times). Consecutive patients referred to an endodontist service from the General Dental Practitioners were invited to participate in the present study if they met the following inclusion criteria: age > 18 years old and a treatment plan including a non-surgical endodontic (both primary Endodontic treatment and retreatment). The treatments were carried out from February 2014 to March 2019 until a total of 101 patients were recruited. If more than one tooth needed root canal treatment in the same patient, only the first treatment conducted was included in the present study. In addition, the number of previous root canals treatment of each patient was recorded. The total sample size was calculated based on a previous pilot study conducted under the same premises.

### 2.2. Procedure

After taking the medical, sociodemographic data (age and sex) and previous dental history, extraoral and intraoral exams were carried out. Also, a preoperative periapical radiography of the tooth to be treated was taken. All data were recorded by one researcher who is the same person who performed all root canal treatments. Only patients diagnosed with any pulp or periapical pathology that required non-surgical endodontic treatment were invited to participate in the present study.

Root canal treatments were performed following the recommendations of the European Society of Endodontology [47]. Following local anesthesia, teeth were isolated using rubber dam and an access cavity preparation was made with a diamond bur high speed size 014 (Komet^®^, Lemgo, Germany). The chemomechanical protocol was carried out using 5.25% sodium hypochlorite between each file used, and the shaping carried out with a combination of with hand (k-files, Denstply Maillefer^®^, Baillagues, Switzerland) and rotary files (ProTaper Universal files, Denstply Maillefer^®^, Baillagues, Switzerland). A final rinse of 18% EDTA (Ultradent^®^, St Louis, MO, USA) was used. Finally, canals were dried using paper points and obturated, using AH Plus (Denstply Sirona^®^, Baillagues, Switzerland) and gutta-percha using a continuous wave down pack technique of obturation (System B, SybronEndo^®^, Glendora, CA, USA).

### 2.3. Measures

#### 2.3.1. Pain Intensity during and after Endodontic Treatment

A numerical rating scale (NRS) was used to assess sensory pain level. This scale was introduced in 1978 by Downie [48]. This scale is among the most commonly used measures of pain intensity in clinical and research settings due to its validity and sensitivity [49]. Patients had to circle a number between 0 (no pain) and 10 (maximum pain). In our study, this scale was administrated at two time points: (1) during treatment at cone fit radiography and (2) at appointment of final restoration one week after treatment.

#### 2.3.2. Pain Anticipation

Was used an item ad hoc by using the question: How do you feel the pain will be during treatment? The measure uses a 10-point Likert scale from 0 = “no pain” to 10 = “maximum pain”. This measure was registered at waiting room before treatment (Figure 1).

#### 2.3.3. Self-Efficacy

To assess self-efficacy, the Spanish version of the ‘General Self-efficacy Scale’ was used [50]. The scale is a brief and widely used instrument to explain and predict human characteristics in different domains, including health behaviors.

This scale is composed of 10-items such as “I can always manage to solve difficult problems if I try hard enough”, “If someone opposes me, I can find the means and ways to get what I want” or “If I am in trouble, I can usually think of a solution”. The scale is scored on a 4-point Likert-type response format ranging from 1 (strongly disagree) to 4 (strongly agree), with a range of total scores from 10 to 40. The higher the score obtained on the scale, the higher the level of the patient’s self-efficacy. It was registered at baseline in the waiting room to evaluate the patient´s belief in their ability to control their symptoms. The internal consistency of this scale was high (Cronbach´s alpha = 0.83) in the present study.

#### 2.3.4. Avoidance during and after Endodontic Treatment

Were assessed using two items *ad hoc*. Both them uses a 10-point Likert scale from 0 “no avoidance” to 10 = “maximum avoidance”. To evaluate the avoid level towards the treatment during the treatment (at cone fit X-ray) patients answered the question: “To what extent would you avoid this situation?”. One week after the root canal treatment was completed, the patients had to assess their degree of avoidance they recalled during the endodontic treatment answered the question: “To what extent would you still avoid having a Root canal treatment?”.

### 2.4. Data Analysis

First, descriptive and bivariate Pearson correlation analyses were performed. For the analysis of pain evolution, Student’s t-analysis for related samples was carried out. Next, a series of multivariate regressions were computed with the PROCESS macro (model 1) [51]. In each regression, a combination of the independent variable (i.e., pain anticipation), the moderator (i.e., self-efficacy) and their interaction was entered to predict the outcome (i.e., pain perception or avoidance). These analyzes were also carried out considering both outcomes both during and after treatment. In total four multivariate regressions were performed. Post hoc analyses were computed when a significant moderation was found to obtain the conditional effects of the independent variables on outcomes at different levels of the moderator. An alpha level of 0.05 was set for all analyses. All analyses were conducted with SPSS version 22 (IBM Corp, Armonk, NY, USA, 2013) [52].

## 3. Results

Two participants were excluded from the present study as their endodontic treatment could not be completed and teeth had to be extracted due to non-restorable root fractures. The final sample size was composed of 99 patients ranged in age from 18 to 72 (mean, 42.91 years ± 11.9 SD). A total of 60% of those were women.

Of the total sample, 74% (*n* = 99) of the subjects had undergone at least one previous root canal treatment. Forty-six percent of treated teeth were molars; 29 percent were premolars, 7 percent were canines, and 17 were incisors (Figure 2). 31 percent were diagnosed with needed retreatment and 68 percent were primary endodontic treatment.

### 3.1. Means, Standard Deviations and Pearson Correlations between Study Variables

Means, standard deviations, and Pearson correlations between study variables are presented in Table 1. Pain during treatment was positively associated with pain after treatment (*p* < 0.001), avoidance during treatment (*p* < 0.001), avoidance after treatment (*p* < 0.001), and pain anticipation (*p* < 0.001). Pain after treatment was positively correlated with avoidance during treatment (*p* < 0.001), avoidance after treatment (*p* < 0.001), and pain anticipation (*p* < 0.001). Avoidance after treatment was positively correlation with pain anticipation (*p* < 0.001). Self-efficacy does not present a positive correlation with pain during and after treatment, avoidance during and after treatment, or with pain anticipation.

### 3.2. Evolution of Pain Intensity

The intensity of pain registered during the treatment was 2.19 ± 2.30 and the mean value of pain registered one week after the treatment (appointment of final restoration) was 1.88 ± 2.16. The difference of means during versus after was significant (*t* = 3.127, *p* = 0.002).

### 3.3. Influence of Pain Anticipation and Self-Efficacy on Pain Perception/Intensity during and after Endodontic Treatment

The results of the regression analyses, including the analysis of moderation of self-efficacy, are presented in Table 2. The results show that the prediction of pain during and after endodontic treatment were only explained by pain anticipation (*p* ˂ 0.001). This influence is positive: the more pain anticipation, the higher pain experience during treatment (Beta = 0.51, *t* = 5.82, *p* ˂ 0.001, 95%CI = 0.34, 0.69) and in the same way, the more pain anticipation, higher pain registered after treatment (Beta = 0.38, *t* = 4.35, *p* ˂ 0.001, 95%CI = 0.21, 0.55). Pain anticipation plays a role more relevant during treatment than after treatment.

Self-efficacy did not influence pain intensity registered during and after endodontic treatment.

### 3.4. Influence of Pain Anticipation and Self-Efficacy on Avoidance during and after Endodontic Treatment

In Table 3, we have presented the results of the regression analyses, including the analysis of moderation of self-efficacy. Pain anticipation had a relevant and fundamental role in avoidance during endodontic treatment (*p* ˂ 0.001) with a positive influence (Beta = 0.51, *t* = 3.60, *p* ˂ 0.001, 95%CI = 0.23, 0.80). Also, pain anticipation was significantly related to the prediction of the avoidance after treatment (*p* ˂ 0.001) with a positive relationship (Beta = 0.62, *t* = 4.29, *p* ˂ 0.001, 95%CI = 0.33, 0.91). Self-efficacy had a significant role in avoidance during endodontic treatment (*p* = 0.03) so that given a higher level of self-efficacy, there was also a higher level of avoidance during treatment (Beta = 0.12, *t* = 2.19, *p* = 0.03, 95%CI = 0.01, 0.23). Self-efficacy was not significant respect to avoidance after treatment.

Table 4 describes the conditional effects of levels of self-efficacy in the relationship between pain anticipation and avoidance during treatment. It was observed a strong relationship between pain anticipation and avoidance when the patient´s self-efficacy levels were moderate (Beta = 0.44, *t* = 3.24, *p* = 0.002, 95%CI = 0.17, 0.72) or high (Beta = 0.84, *t* = 3.5, *p* = ˂ 0.001, 95%CI = 0.37, 1.33), and this moderation effect is represented in Figure 3. In individuals with high levels of self-efficacy, pain anticipation will directly and significantly predict patient´s avoidance. In subjects with lower levels of self-efficacy, pain anticipation is independent of avoidance during treatment.

## 4. Discussion

Perception of pain is multifactorial, and the expression of pain is a legitimate report of the biology and psychology of a person [53,54]. Therefore, knowing more about the psychological aspects of a patient and how they face a dental procedure that is associated with pain and their anticipation would allow dentists to improve their level of care and, consequently, increase patient´s confidence in the professional. Not only the individual perception regarding the treatment but also the negative social connotations associated to the treatment has an influence in this respect. The main objective of performing a root canal treatment is the maintenance of the tooth. Nonetheless, it is commonly known by the patients as “killing the nerve” in Spain. Therefore, in the popular imagination, the performance of root canal treatment is frequently associated with pain [55,56].

In the present research, pain anticipation was significantly higher than pain registered during the procedure and one week after completion. These results are in accordance with the previously published literature [13,41,44,55,57,58,59]. The novelty of the present study relies on using the variable “pain anticipation” as an independent variable that predicts pain evolution over time and patient’s avoidance behavior. Pain anticipation modulates perceived pain during and after endodontic treatment, this being more relevant during the procedure.

The role of self-efficacy and the possibility that self-efficacy could be applied in dentistry was also assessed in the present study. Some authors indicate that depending on their perception of the event, people can experience very different emotional responses to very similar levels of stimuli intensity [60,61]. Therefore, self-efficacy has an essential role as a mediator of the individual’s perception of the situation that anticipates as stressful. But relatively few articles have been focused on patient experience. Up to this date, we have not found any paper that evaluates the levels of self-efficacy before an endodontic therapy. In the present study, self-efficacy has an insignificant role in perceived pain.

Nevertheless, in terms of avoidance, pain anticipation has an essential role before and after the treatment. The relation between pain anticipation and avoidance has already been established [13,58]. As for the analysis of the influence of pain anticipation over avoidance, this was is stronger during endodontic treatment than after. This relationship was more significant with respect to avoidance than perceived pain.

The evidence suggests that how a person perceives the dental environment is a considerably more important determinant of avoidance than having had a previous distressing experience at a dental visit [2,62]. However, the authors have not found any references that study patient’s self-efficacy as a variable that moderates pain before an endodontic treatment. It stands out that Kent [63] already suggested that self-efficacy can be applied to the control of symptoms of anxiety in dentistry in the 1980s. However, this concept has not been developed.

The results of the present study showed that self-efficacy had a significant role in terms of avoidance during endodontic treatment, thus the importance of further study the role of self-efficacy in dentistry. This is due to the fact that patients showing a higher level of self-efficacy. They also presented an increased avoidance behavior. Due to the adaptive role of self-efficacy in health results [64,65,66] and in oral and dental health in particular [67,68], these results might seem contradictory. However, there are no previous publications in the field of endodontics so the interpretation of the present results is complex. A possible explanation for this would be that self-efficacy is linked to the patients’ need for control. Nevertheless, in a dental setting patients may experience perceptions of uncontrollability, unpredictability and dangerousness (cognitive vulnerability [69]). Patients with high levels of self-efficacy need to have control over the situations. It is difficult for them to have to delegate those capabilities to the dentist. In the present study, self-efficacy measured after treatment was not significant this might be explained by the generation of trust by patient towards the dentist so patients did not value the dental situation so adversely. For that reason, further research in this field is necessary, in particular regarding the new index of trust in dentists [70].

Self-efficacy had an essential role as a moderator of pain anticipation and avoidance during treatment. The relationship between pain anticipation and avoidance has already been established in the literature [13,58,71]. Additionally, our results show that the aforementioned relationship is not universal for all patients and that it is more relevant for patients showing moderate or high levels of self-efficacy. As there are no previous similar results to ours, it could be hypothesized that pain anticipation could be a more significant stressor in those patients showing high self-efficacy. In order to create a successful patient-practitioner relationship, dental practitioners should first identify individuals presenting moderate or high levels of self-efficacy and then adopt an appropriate tailored approach for the specific patient´s concerns.

This study presents some limitations that must be taken into account. First, the sample size is small and was collected at two different sites. Therefore the findings cannot be generalized. Some authors explain that the level of expertise of the operator can have an influence in the results [36,44,68]. However, in the present study there was only one experienced endodontist involved. In the present study, only one clinical protocol was used for all the patients. However, some authors have already showed differences in terms of pain after using different shaping files, irrigation protocols or sealers [11,12,72,73]. Finally, it should be mentioned the possible bias in the present study due to 74% of the patients had undergone previous endodontic treatment. As we have already commented, endodontic treatment is a very common treatment in the adult population, and the present study represents the oral health of the Spanish population faithfully. Although this bias can be minimized to avoid a preconceived notion of pain or avoidance to procedure, the influence of this variable should be studied in future research.

## 5. Conclusions

Despite the limitations of this study, the most novel result of our research is the role that self-efficacy plays in the relationship between pain anticipation and avoidance during endodontic treatments, especially in patients with high levels of self-efficacy. The relationship between pain anticipation and avoidance was already well documented in previous studies [62,74,75,76]. The present work shows the importance of taking into account self-efficacy as a variable that can contribute positively to avoidance during treatment. This has important clinical repercussions that point at the need to assess patient’s self-efficacy before conducting a root canal treatment using very simple scales such the ones presented here (General Self-Efficacy Scale with 10 items). This evaluation could be carried out together with the signing of the informed consent.

The present findings show the need to take into account patient’s self-efficacy in order to optimize the results of the procedure and reduce its avoidance. The specialist could provide positive information to the patient about the treatment. In particular, if the patient shows high levels of self-efficacy, it would be interesting to work particularly with very specific instructions on how the procedure is going to go, the duration of the appointment, what the patient will feel during the treatment, and help them to solve all of their doubts. These explanations would guarantee that the need for control of the patient with high levels of self-efficacy is satisfied with the trust in the dentist delegating control to the professional.

An individual approach should always be taken into account for each patient, and all clinical staff should be taught to follow these recommendations. This may help dentists adopting perceptions of self-efficacy respect to treatment they conduct. Also, it could reduce the risk of patients failing their dental appointments and its consequent deteriorated oral health.

## Figures and Tables

**Figure 1 ijerph-19-01399-f001:**
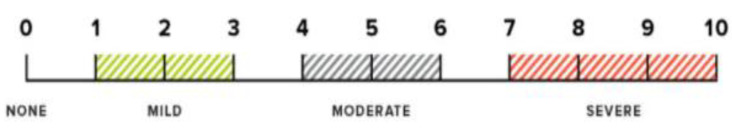
How do you feel the pain will be during treatment?

**Figure 2 ijerph-19-01399-f002:**
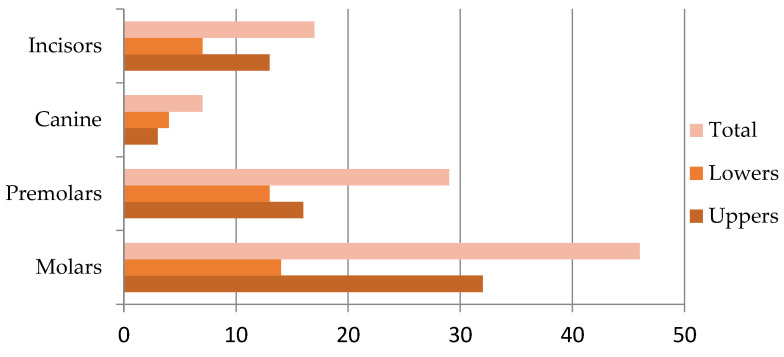
Distribution of treated teeth by tooth type (*n* = 99).

**Figure 3 ijerph-19-01399-f003:**
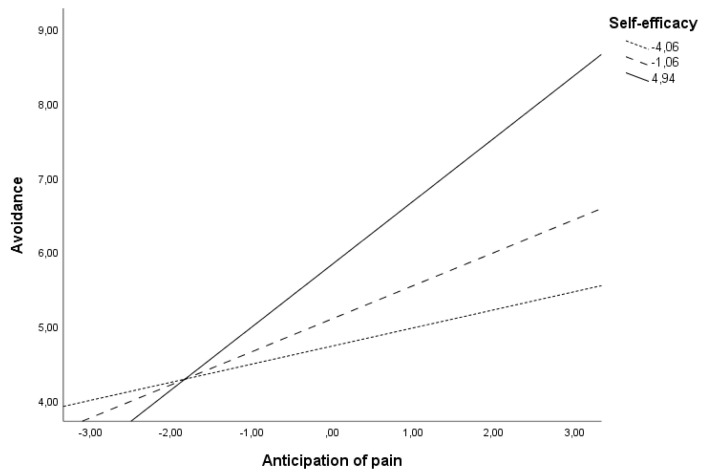
Relationship between anticipation of pain and avoidance at different levels of self-efficacy.

**Table 1 ijerph-19-01399-t001:** Means, standard deviations, and Pearson correlations between study variables (*n* = 99).

	Mean (SD)	2. Pain after Treatment	3. Avoidance during Treatment	4. Avoidance after Treatment	5. Pain Anticipation	6. Self-Efficacy
1. Pain during treatment	2.19 (2.30)	0.903 **	0.321 **	0.355 **	0.530 **	−0.093
2. Pain after treatment	1.87 (2.16)		0.372 **	0.393 **	0.436 **	−0.029
3. Avoidance during treatment	5.08 (3.42)			0.869 **	0.283 **	0.082
4. Avoidance after treatment	4.75 (3.53)				0.386 **	0.092
5. Pain anticipation	4.11 (2.40)					−0.138
6. Self-efficacy	32 (6.86)					

Note. Significance levels ** *p* < 0.01; SD, standard deviation.

**Table 2 ijerph-19-01399-t002:** Prospective prediction of pain during and after treatment from anticipation of pain, self-efficacy and their interaction.

	*R* ^2^	*F*	Beta	*t*	*p*	95% CI
DV = Pain during treatment	0.28	12.43 ***				
Pain anticipation			0.51	5.82	<0.001	0.34, 0.69
Self-efficacy			<0.001	−0.04	0.970	−0.07, 0.07
Interaction			0.006	0.33	0.745	−0.03, 0.04
DV = Pain after treatment	0.19	7.60 ***				
Pain anticipation			0.38	4.35	<0.001	0.21, 0.55
Self-efficacy			<0.001	0.004	0.996	−0.07, 0.07
Interaction			−0.01	−0.53	0.596	−0.05, 0.03

Note. Significance levels *** *p*< 0.001. DV, dependent variable; *R*^2^, coefficient of determination; *F*, *F*-statistic in linear regression; Beta, beta coefficient in multiple regression; *t*, *t*-test in linear regression; *p*, *p*-value; 95%CI, 95% confidence interval.

**Table 3 ijerph-19-01399-t003:** Prospective prediction of avoidance during and after treatment from anticipation of pain, self-efficacy, and their interaction.

	*R* ^2^	*F*	Beta	*t*	*p*	95% CI
DV = Avoidance during treatment	0.14	4.96 **				
Anticipation of pain			0.51	3.60	<0.001	0.23, 0.80
Self-efficacy			0.12	2.19	0.030	0.01, 0.23
Interaction			0.07	2.11	0.037	0.004, 0.129
DV = Avoidance after treatment	0.17	6.64 ***				
Anticipation of pain			0.62	4.29	<0.001	0.33, 0.91
Self-efficacy			0.093	1.65	0.100	−0.01, 0.21
Interaction			0.02	0.63	0.531	−0.04, 0.08

Note. Significance levels ** *p* < 0.01; *** *p* < 0.001. DV, dependent variable; *R*^2^, coefficient of determination; *F*, *F*-statistic in linear regression; Beta, beta coefficient in multiple regression; *t*, *t*-test in linear regression; *p*, *p*-value; 95%CI, 95% confidence interval.

**Table 4 ijerph-19-01399-t004:** Conditional effects of anticipation of pain on avoidance at values of self-efficacy.

Self-Efficacy	Beta (Anticipation of Pain)	*t*	*p*	95% CI
−4.06	0.243	1.512	0.133	−0.08, 0.56
−1.06	0.445	3.248	0.002	0.17, 0.72
4.93	0.847	3.503	<0.001 ***	0.37, 1.33

Note. Significance levels *** *p* < 0.001; Beta, beta coefficient in linear regression; *t*, *t*-test in linear regression; *p*, *p*-value; 95%CI, 95% confidence interval.

## Data Availability

The data presented in this study are available on request from the corresponding author. The data are not publicly available due to privacy restrictions.

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
