# Peer review of "Pain and Avoidance during and after Endodontic Therapy: The Role of Pain Anticipation and Self-Efficacy"

_ijerph, 2022, doi:10.3390/ijerph19031399_

Round 1
Reviewer 1 Report
Thank you for submitting your manuscript for review.
This is an interesting topic and your research provides insight into how best to treat our patients.
General comments:
- Overall, please review the manuscript for English grammatical errors and sentence structure. In its current format, there are sentences that are hard for the reader to interpret due to grammar and sentence structure errors.
- References, please make sure to place commas after each reference number in the text of the manuscript between []
Introduction:
- I encourage you to explain self-efficacy further. once the reader gets to the results and discussion the idea of self-efficacy is confusing as there is not a strong definition. Maybe provide an example in the paragraph lines 45-54 in a dental context or from a previous paper. This would help the reader grasp the concept.
- You clearly state your aim of the study
Methods:
- In the first paragraph, lines 83-86, I would recommend that you mention the procedures were completed by one endodontist (from lines 111-113).
- With subjects included that have had previous RCT on the same tooth (re-treatment), do you think this invited bias regarding the procedure from the patient? Of the 99 subjects, how many had previous RCT and thus have a preconceived notion of pain or avoidance to the procedure? These could also be listed as limitations of potential bias in the study.
- You mention the Likert scale for pain anticipation and separate scale for self-efficacy (4 point scale consisting of 10 items). It would be helpful for the reader to view these questions and scales as a figure. This will also help to explain tables 2-4 and graph 1.
- Graph 1 should be relabeled "Figure 2"
Results:
- In tables 1-4 please add additional information to explain the table material. For example in table 1, stating for the reader what 2,3,4,5,6 mean, In table 2, state what R2 and F, etc. stand for. This will help the reader understand your results.
Discussion:
- The results of a higher level of self-efficacy assoc. with a higher level of avoidance during treatment (line 199-200) seems to be opposite of your definition in the introduction. You do mention this contradiction in your discussion. but it could be helpful to expand on why you think this is the case further in your discussion.
Conclusions:
1. Line 289-292, you state "the most novelty result of our research is the roll that self-efficacy plays in the relationship between pain anticipation and avoidance during endodontic treatment, especially in patients with high levels of self efficacy. This relationship was already well documented in previous studies." What relationship are you stating is well documented? Pain anticipation and avoidance or self-efficacy and avoidance or pain anticipation? It seems this study uniquely finds that there is a relationship between self-efficacy and avoidance that is newly found in this study. Please add references to support previous studies.
Author Response
General comments:
- Overall, please review the manuscript for English grammatical errors and sentence structure. In its current format, there are sentences that are hard for the reader to interpret due to grammar and sentence structure errors.
- References, please make sure to place commas after each reference number in the text of the manuscript between []
Thanks for your time dedicated to the correction of this paper. Hope all grammatical mistakes and errors in the structure of the sentences have been corrected. Also, all references were checked.
Introduction:
- I encourage you to explain self-efficacy further. once the reader gets to the results and discussion the idea of self-efficacy is confusing as there is not a strong definition. Maybe provide an example in the paragraph lines 45-54 in a dental context or from a previous paper. This would help the reader grasp the concept.
Thank you very much for your suggestion. We have deepened the self-efficacy ‘concept in order to clarify the results of this article.
Methods:
- In the first paragraph, lines 83-86, I would recommend that you mention the procedures were completed by one endodontist (from lines 111-113).
Thanks for your recommendation, it is a more logical order to mention in the first paragraph that all the treatments were done by one endodontist. It is a great idea unifies this paragraph.
- With subjects included that have had previous RCT on the same tooth (re-treatment), do you think this invited bias regarding the procedure from the patient? Of the 99 subjects, how many had previous RCT and thus have a preconceived notion of pain or avoidance to the procedure? These could also be listed as limitations of potential bias in the study.
Thanks to your correction, we have incorporated the number of previous endodontic treatments that the patients had undergone. This variable had not been previously taken into account but it is very important and has been considered as a possible bias of the study.
- Results:
- In tables 1-4 please add additional information to explain the table material. For example in table 1, stating for the reader what 2,3,4,5,6 mean, In table 2, state what R2 and F, etc. stand for. This will help the reader understand your results.
We have incorporated as much information as possible in the tables to make easier the interpretation of the results.
- Discussion:
- The results of a higher level of self-efficacy assoc. with a higher level of avoidance during treatment (line 199-200) seems to be opposite of your definition in the introduction. You do mention this contradiction in your discussion. but it could be helpful to expand on why you think this is the case further in your discussion.
We understand that our result is complex, especially due to the lack of previous papers available to interpret it. For that reason, the paragraph was modified as follows: “A possible explanation for this would be that self-efficacy is linked to the patients’ need for control. Nevertheless in a dental setting, patients may experience perceptions of uncontrollability, unpredictability and dangerousness (Cognitive vulnerability [69]). Patients with high levels of self-efficacy need to have control over the situations. It is difficult for them to have to delegate those capabilities to the dentist.”
- Conclusions:
- Line 289-292, you state "the most novelty result of our research is the roll that self-efficacy plays in the relationship between pain anticipation and avoidance during endodontic treatment, especially in patients with high levels of self efficacy. This relationship was already well documented in previous studies." What relationship are you stating is well documented? Pain anticipation and avoidance or self-efficacy and avoidance or pain anticipation? It seems this study uniquely finds that there is a relationship between self-efficacy and avoidance that is newly found in this study. Please add references to support previous studies.
Thank you for highlighting that relationship, as prior to your review it was unclear. We have improved the concept and we have relied on published literature. The paragraph has been modified adding the importance of the relationship between anticipation of pain and avoidance, especially in patients with high levels of self-efficacy. This is our most important finding.

Reviewer 2 Report
Manuscript Title: PAIN AND AVOIDANCE DURING AND AFTER eNDODONTIC tHERAPY. tHE ROLE OF PAIN ANTICIPATION AND SELF-EFFICACY.
Overview: analyze the perception of pain during endodontic therapy, at two diferente time points:
a)during the procedure
b)at the end of the treatment
There are also two additional objectives
- To analyze the posible influence of pain articipation and self-efficacy in the perception of pain during and after the treatment
- To asses the posible influence of pain anticipation and self-efficacy in the avoidance during and after treatment.
General comments: This is an interesting manuscript that addresses an important area of unmet medical need. Please see my specific comments below for more details.
This manuscript consists of a non-structured abstract with 4 keywords, 5 sections (introduction, materials & methods with 4 subsections, results with 4 subsections, discussion with Limitations of the study, and conclusions) on 11 pages of single-spaced text with embedded figures and tables. There are 72 references,1 figure, 4 tables, and 1 graph.
Specific comments:
- The keywords are absolutely fine.
- Introduction: Lines34,35,57,61,63,66…230 check writing references in all the manuscript [23456] = [2-6]
The therm “self-efficacy” it is posible to use acronym? Is repeat many times will make reading easier.
- Materials and methods: Line 93/94 it is posible add reference?
Author Response
Thank you so much for the time dedicated to the correction of this paper. All references were checked and some has been added.
I am so sorry to inform you that we have not found any acronym for self-efficacy.
Thanks again
